



# Wind wave and water level dataset for Hornsund, Svalbard (2013-2021)

Zuzanna M. Swirad[1], Mateusz Moskalik[1], Agnieszka Herman[2]

[1]Institute of Geophysics, Polish Academy of Sciences, Warszawa, Poland
[2]Institute of Oceanology, Polish Academy of Sciences, Sopot, Poland
*Correspondence to*: Zuzanna M. Swirad (zswirad@igf.edu.pl)

**Abstract.** Underwater pressure sensors were deployed at various locations of the nearshore (8−23 m depth) Hornsund fjord, Svalbard between July 2013 and February 2021. Raw pressure measurements at 1 Hz were used to derive mean water levels, wave spectra and bulk wave parameters for 1024 s bursts at hourly intervals. The procedure included subtracting atmospheric pressure, depth calculation, Fast Fourier Transform, correction for the decrease of the wave orbital motion with depth and adding a high-frequency tail. The dataset adds to the sparse in situ measurements of wind waves and water levels in the Arctic, and can be used e.g. for analysing seasonal wind wave conditions and inter-annual trends, and calibrating/validating wave models.

## 1 Introduction

In situ wave measurements are critical for understanding wave climate, analysing seasonal and inter-annual trends, and calibrating and validating wave transformation models (e.g. Reistad et al., 2011). Spatial distribution of instruments providing wind wave information is irregular and tends to concentrate in mid- and low-latitude coastal areas (e.g. https://www.ndbc.noaa.gov/; Semedo et al., 2015). In the Arctic, the network of such instruments is particularly sparse. There is a pertinent lack of continuous wave data in Svalbard archipelago where communities, industry infrastructure, and research stations are located. Continuous wave observations in the coastal Arctic are needed to better understand how decreasing sea-ice extent (Barnhart et al., 2014; IPCC, 2019), increasing storminess (Francis et al., 2011; Wang et al., 2015; Stopa et al., 2016; Waseda et al., 2018), and, in consequence, larger waves acting on Arctic coasts for longer time periods contribute to coastal flooding and erosion, that can cause infrastructure damage (Forbes, 2011).

Our knowledge of the western Svalbard wave climate comes primarily from global spectral models such as NOAA's WaveWatch III (WW3) hindcast (WW3DG, 2019), ECMWF reanalysis projects ERA-40 (1957-2002; Uppala et al., 2005), ERA-Interim (1979-2019; Dee et al., 2011) and ERA5 (1959-present; Hersbach et al., 2020), or NCEP's Climate Forecast System Reanalysis (CFSR; Saha et al., 2014). Arctic Ocean Wave Analysis and Forecast system (Carrasco et al., 2022) is a shorter duration (since 2017), higher resolution (3 km) model that provides e.g. significant wave height ($H_s$), peak period ($T_p$) and peak wave direction ($\theta_p$) hourly using ECMWF's WAM model. The 10 km resolution ERA-40 reanalysis allowed Semedo et al. (2015) to capture seasonal trends in swell vs seas dominance and the ≥ 10 cm per decade increase in winter $H_s$ over the northern Atlantic. Stopa et al. (2016) used CFSR and altimetry data to calculate average $H_s$ of 1.5 m (99[th] percentile of 5-6 m) for the period 1992-2014 west of Svalbard. Wojtysiak et al. (2018) observed up to 1 m $H_s$ difference between winter (higher) and summer (lower) months using WW3 (2005−2015; at 0.5º resolution) and ERA-Interim (1979−2015; at 1º resolution), and found a statistically-significant trend of increasing frequency (2 storms per decade) and total duration (4 days per decade) of storms for the Greenland Sea off south-western Svalbard for the 1979−2015 period, with the typical annual values of 10-40 storms and 20-80 days, respectively.

Herman et al. (2019) used three nested Simulating Waves Nearshore (SWAN; Booij et al., 1999) models to predict wind wave parameters within bays of Hornsund fjord (~15 m depth) taking eastern Greenland Sea WW3 spectra as boundary conditions.



They ran the model for two sea-ice free 4-month periods (August – November 2015 and 2016) finding a good agreement between the modelled and measured total wave energy ($r^2 > 0.9$) and wave period ($r^2 = 0.63–0.78$) (Herman et al., 2019).

The large-scale models are good for understanding the general trends in the Arctic/Svalbard area, but provide limited information on local-scale wave parameters in specific fjords and bays (Nederhoff et al., 2022). How the open ocean wave conditions translate into wave conditions in the coastal areas is poorly constrained given complex coastal wind patterns and bottom topography (Semedo et al., 2015). Moreover, the large-scale models over-simplify most aspects of wind wave-sea ice interactions. Most operational models use simple empirical formulae for wave attenuation in sea ice (Barnhart et al., 2014;

Zhao et al., 2015; Ardhuin et al., 2016). The study of Herman et al. (2019) added a considerable detail into wind wave transformation in the nearshore environment of Hornsund. However, the model tested against buoy data performed well for ice-free conditions only. For a bay of Beaufort Sea, Nederhoff et al. (2022) incorporated sea ice into SWAN model which enabled to reliably describe wave climate in 1979–2019. The need for observational data to validate wave models, especially in periods when the sea ice is present, persists.


We present a 7.5-year (2013-07 to 2021-02) wind wave dataset from Hornsund, southern Svalbard. Our goal is to increase observational understanding of Arctic wave conditions by providing a dataset that can be further used to e.g. i) analyse the inter- and intra-annual trends in nearshore wind wave conditions, ii) calibrate and validate wave transformation models, iii) quantify the role of sea ice in wave attenuation, iv) create empirical models of wave run-up on high-latitude beaches, and, v)

predict future wind wave conditions.

**2 Study area**

Hornsund is a ~30 km long fjord of SW Spitsbergen, Svalbard (**Fig. 1a**). It has a ~12 km wide and ~100 m deep opening to the Greenland Sea. The average fjord depth is ~100 m with the deeper (200-250 m) central part (**Fig. 1b**; Herman et al., 2019). The tides are semi-diurnal and the average tidal range is 0.75 m (Kowalik et al., 2015). The circulation is cyclonic (counter-

clockwise) with the inflow from SW and outflow to the NW (Jakacki et al., 2017).

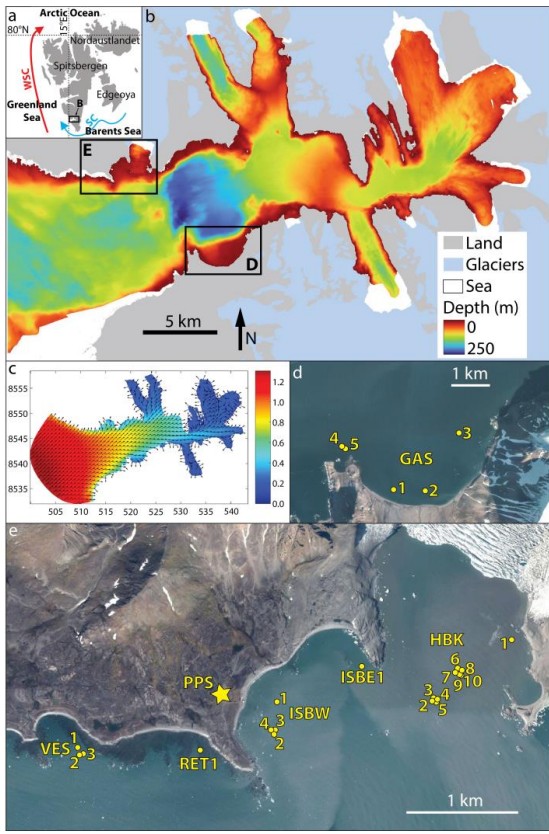

**Figure 1: Study area: (a) Svalbard archipelago; WSC = warm West Spitsbergen Current; SC = cold Sørkapp Current; (b) bathymetry of Hornsund fjord (source: Norwegian Hydrographic Service; permit granted to IG PAS); (c) mean significant wave height, $H_s$ (colours, in m) and wave direction, $\theta_p$ (arrows) from Herman et al. (2019); axis labels refer to UTM33X coordinates (in km); location of sensor deployments in southern (d) and northern (e) Hornsund. HBK = Hansbukta, ISB = Isbjørnhamna (W = western, E = eastern), RET = Rettkvalbogen, GAS = Gåshamna, VES = Veslebogen, PPS = Polish Polar Station.**

In 1979−2018 easterly winds dominated at the Polish Polar Station (12 m a.s.l.; PPS in **Fig. 1e**) with the mean direction of 124º (annual mean range of 102-140º). Mean wind speed at ~20 m a.s.l. was 5.5 m s$^{-1}$ (Wawrzyniak and Osuch, 2020).

Wave conditions in Hornsund are usually related to the long oceanic swell or mixed swell/wind sea from S−SW with short wind waves formed locally due to predominantly easterly winds. The mean $H_s$ at the fjord mouth is 1.2−1.3 m decreasing to 0.5−0.9 m in the central and to < 0.4 m in the inner parts of Hornsund (**Fig. 1c**). Northern shores of the fjord receive more wave energy than southern shores (Herman et al., 2019).

Hornsund bays (in this study Hansbukta, Isbjørnhamna, Rettkvalbogen, Veslebogen and Gåshamna) have complex shapes and bottom topography with ubiquitous skerries causing strong wave transformation due to refraction and dissipation (Herman et al., 2019).

Sea ice forms locally in the fjord or drifts from the open Greenland Sea. The latter originates east of Svalbard, drifts past the southern tip of Spitsbergen (Sørkapp) and then northwards along the western Spitsbergen coast with cold Sørkapp Current



(blue arrow in **Fig. 1a**). Fast ice (i.e. sea ice attached to the shore) persists during winter months. Muckenhuber et al. (2016) observed a decrease in sea ice (both drift and fast ice) duration and extent between 2000 and 2014. In summer months glacier

ice from calving tide-water glaciers (Błaszczyk et al., 2019) may accumulate in bays. Increased storminess coincident with positive air temperature anomalies and the lack of sea ice, in particular in October–December, may contribute to coastal erosion (Zagórski et al., 2015).

## 3 Methods

### 3.1 Input data

Pressure data were collected between 2013-07-21 and 2021-02-12 using RBR virtuoso P (continuous sampling at 4 or 6 Hz interval), RBR duo TD (continuous sampling at 1 Hz interval) and RBR virtuoso wave (1024 s bursts at 30 min interval with 1 Hz sampling interval or at 60 min interval with 2 Hz sampling interval). There were 24 single deployments with duration of 13−599 days (**Table 1**; **Fig. 2**). The instruments were anchored to the sea bottom in various locations in northern (Hansbukta, western and eastern Isbjørnhamna, Rettkvalbogen, Veslebogen) and southern (Gåshamna) Hornsund (**Fig. 1d,e**). The raw

pressure data are part of the LONGHORN oceanographic monitoring of IG PAS and are provided in Swirad et al. (2022).



**Table 1: Details of the pressure sensor deployments for in situ wave measurements in Hornsund, Svalbard. Deployment ID (DepID) refers to bays: HBK = Hansbukta, ISB = Isbjørnhamna (W = western, E = eastern), RET = Rettkvalbogen, GAS = Gåshamna, VES = Veslebogen. LONGHORN ID refers to the IG PAS oceanographic monitoring (Swirad et al., 2022).**

| DepID | LONGHORN ID | Start | End | Length (days) | X (m UTM33X) | Y (m UTM33X) | Depth (m) | Instrument, serial number |
|---|---|---|---|---|---|---|---|---|
| HBK1 | P01 | 2013-07-21 | 2013-08-10 | 21 | 516337 | 8547621 | 8 | RBR virtuoso P, 52915 |
| HBK2 | P02 | 2013-09-05 | 2013-12-07 | 94 | 515675 | 8546969 | 23 | RBR virtuoso P, 52915 |
| HBK3 | P03 | 2014-02-01 | 2014-05-05 | 94 | 515675 | 8546969 | 23 | RBR virtuoso P, 52916 |
| HBK4 | P04 | 2014-06-01 | 2014-09-02 | 94 | 515681 | 8546960 | 23 | RBR virtuoso P, 52915 |
| HBK5 | P05 | 2014-08-25 | 2014-11-26 | 94 | 515681 | 8546960 | 23 | RBR virtuoso P, 52916 |
| HBK6 | Wave01 | 2015-06-10 | 2016-06-02 | 359 | 515812 | 8547208 | 22 | RBR virtuoso wave, 52980 |
| HBK7 | Wave04 | 2016-07-01 | 2017-05-21 | 325 | 515799 | 8547208 | 22 | RBR virtuoso wave, 52980 |
| HBK8 | Wave08 | 2017-06-09 | 2018-05-24 | 350 | 515856 | 8547189 | 22 | RBR virtuoso wave, 55113 |
| HBK9 | TD01 | 2018-06-05 | 2019-01-15 | 225 | 515845 | 8547185 | 22 | RBR duo TD, 82445 |
| HBK10 | TD02 | 2018-12-10 | 2019-06-09 | 182 | 515845 | 8547185 | 22 | RBR duo TD, 82446 |
| ISBW1 | P06 | 2015-05-26 | 2015-06-07 | 13 | 514131 | 8546876 | 9 | RBR virtuoso P, 52916 |
| ISBW2 | Wave02 | 2015-06-04 | 2016-06-03 | 366 | 514085 | 8546553 | 10 | RBR virtuoso wave, 55112 |
| ISBW3 | Wave05 | 2016-06-13 | 2017-05-23 | 345 | 514078 | 8546580 | 10 | RBR virtuoso wave, 55112 |
| ISBW4 | Wave07 | 2017-06-03 | 2018-05-22 | 354 | 514061 | 8546579 | 10 | RBR virtuoso wave, 55112 |
| ISBE1 | Wave03 | 2015-06-04 | 2016-06-03 | 366 | 514899 | 8547338 | 10 | RBR virtuoso wave, 55112 |
| RET1 | P07 | 2015-07-13 | 2015-07-25 | 13 | 513334 | 8546193 | 11 | RBR virtuoso P, 52915 |
| GAS1 | P08 | 2015-07-13 | 2015-07-25 | 13 | 520473 | 8540424 | 8 | RBR virtuoso P, 52916 |
| GAS2 | P09 | 2015-08-16 | 2015-09-09 | 25 | 521393 | 8540411 | 8 | RBR virtuoso P, 52915 |
| GAS3 | Wave06 | 2016-06-17 | 2017-06-02 | 351 | 522299 | 8541805 | 11 | RBR virtuoso wave, 55113 |
| GAS4 | Wave10 | 2018-06-05 | 2019-06-10 | 371 | 519433 | 8541387 | 22 | RBR virtuoso wave, 55113 |
| GAS5 | Wave12 | 2019-06-26 | 2021-01-14 | 569 | 519495 | 8541380 | 23 | RBR virtuoso wave, 55113 |
| VES1 | P10 | 2015-08-16 | 2015-09-13 | 29 | 512247 | 8546342 | 11 | RBR virtuoso P, 52916 |
| VES2 | Wave09 | 2018-06-05 | 2019-06-09 | 370 | 512261 | 8546279 | 16 | RBR virtuoso wave, 55112 |
| VES3 | Wave11 | 2019-06-25 | 2021-02-12 | 599 | 512295 | 8546285 | 16 | RBR virtuoso wave, 55112 |

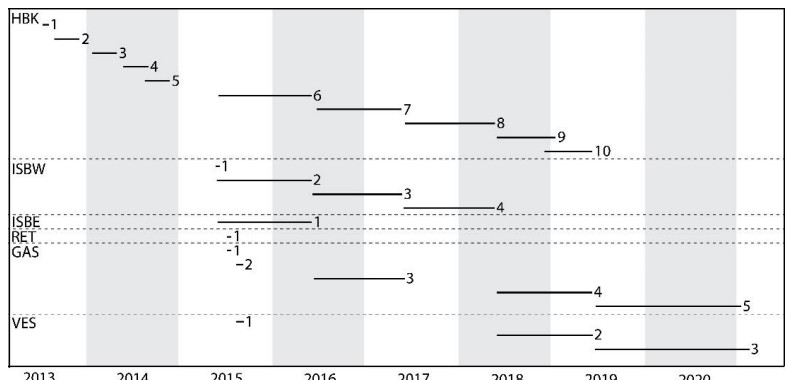

**Figure 2: Timespan of pressure sensor deployments for in situ wave measurements in Hornsund, Svalbard. HBK = Hansbukta, ISB = Isbjørnhamna (W = western, E = eastern), RET = Rettkvalbogen, GAS = Gåshamna, VES = Veslebogen.**



For consistency the raw data were subsampled to 1024 s bursts at 60 min interval (starting at full hours) with 1 Hz sampling interval. The erroneous bursts at the start and end of deployments were removed. The datasets were cropped to full days so that the first measurement occurs at 00:00:00 UTC (hh:mm:ss) and the last one at 23:17:03 (1024[th] s after 11pm). These 24 deployment files are time series with three columns representing time, burst number and raw pressure in dbar, and are available

as part of the dataset (Swirad et al., 2023).

### 3.2 Burst processing

The deployment files were imported into Spyder (Python 3.9) and processed on the burst-by-burst basis. Hourly (one per burst) atmospheric pressure $P_{\mathrm{air}}$ (mbar) at the sea level was taken from the Polish Polar Station archive (https://monitoring-

hornsund.igf.edu.pl/; accessed on 2022-03-28). The water pressure, $P_{\mathrm{sea}}$ (dbar) was calculated by subtracting atmospheric pressure from the raw pressure, $P_{\mathrm{raw}}$:

$$P_{\mathrm{sea}} = P_{\mathrm{raw}} - P_{\mathrm{air}}/100. \tag{1}$$

Depth, $z$ (m) was calculated using UNESCO formula (Fofonoff and Millard, 1983) under assumption of constant water

temperature of 0ºC, salinity of 35 PSU and latitude $\varphi = 77$ºN:

$$z = \left[\left(\left(\left(-1.82 \cdot 10^{-15} P_{\mathrm{sea}} + 2.279 \cdot 10^{-10}\right) P_{\mathrm{sea}} - 2.2512 \cdot 10^{-5}\right) P_{\mathrm{sea}} + 9.72659\right) P_{\mathrm{sea}}\right]/g, \tag{2}$$

where $g$ (m·s$^{-2}$) denotes acceleration due to gravity, computed as:

$$g = 9.780318[1 + (5.2788 \cdot 10^{-3} + 2.36 \cdot 10^{-5} x) x] + 1.092 \cdot 10^{-6} P_{\mathrm{sea}}, \tag{3}$$

and $x$ is given by:


$$x = \sin^2(\varphi/57.29578). \tag{4}$$

The slowly-varying component of water depth (due to, e.g., tide and storm surge) was removed by subtracting from $z$ a least-square-fitted 2$^{\mathrm{nd}}$ order polynomial trend, $z_{\mathrm{lf}}$, resulting in time series $z_{\mathrm{hf}}$ (m), related to depth variability associated with wind waves: $z_{\mathrm{hf}} = z - z_{\mathrm{lf}}$. The energy density spectrum at depth $z$, $E_z(f)$ (in m$^2$·s), was computed in a standard way by applying

Fast Fourier Transform (FFT; Frigo and Johnson, 2005) to the time series $z_{\mathrm{hf}}$.

Finally, the spectrum at the sea surface, $E_0(f)$, was computed from $E_z(f)$ by applying a correction factor $A(f)$ accounting for the decrease of the wave orbital motion (and thus pressure fluctuations) with depth (compare red and blue spectra in **Fig. 3**):

$$E_0(f) = E_z(f)/A(f). \tag{5}$$





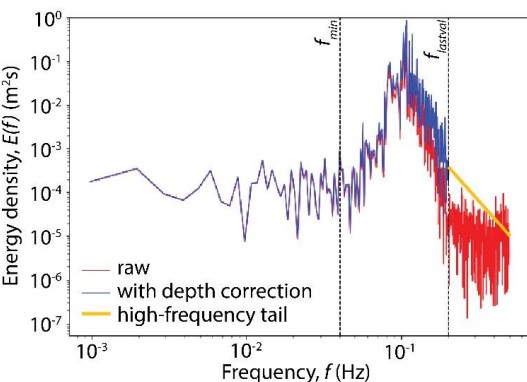

**Figure 3: An example of wave energy density spectrum computed with the algorithm described in the text (deployment HBK9 burst #1): raw spectrum $E_z(f)$ at the depth of the logger (red), depth-corrected spectrum $E_0(f)$ (blue), and the analytical high-frequency tail (yellow). Frequency $f_{min} = 0.04$ Hz is the minimum frequency used to calculate mean wave parameters, and $f_{lastval}$ is the highest frequency reliably measured. The plot is limited to $f = 0.5$ Hz which is the upper limit of the observation data. Wave parameters are calculated in two versions, for $f_{min} < f < f_{lastval}$ and for $f_{min} < f < \infty$.**

To this end, a set $K$ of basic wavenumber values was defined, $K = \{0, 0.01, 0.02, \cdots, 1000\}$ (m⁻¹), and a corresponding set of basic wave frequencies $F$, with elements:

$$f_i = \sqrt{g k_i \tanh(k_i \bar{h})}/(2\pi), \qquad \text{for each } k_i \in K. \tag{6}$$

The set of correction factors $A$ is then given by:

$$A_i = \cosh\left(k_i(\bar{h} - \overline{z_{lf}})\right)/\cosh\left(k_i \bar{h}\right), \qquad \text{for each } k_i \in K, \tag{7}$$

where $\bar{h}$ and $\overline{z_{lf}}$ denote the mean bottom depth and the mean logger depth, respectively (in the present case, with loggers mounted at the bottom, $\bar{h} = \overline{z_{lf}}$; averaging takes place over burst duration). The correction factor in (5) was calculated by linearly interpolating $F$ and $A$ to the frequencies of the energy spectrum. (Note that $g$ in expression (6) was computed from (3, 4) without the last term in (3), i.e., for $P_{sea} = 0$.)

As $A(f)$ quickly decreases with increasing wave frequency, the values of $E_0(f)$ computed from (5) become unreliable for $f$ higher than some limiting frequency $f_{lastval}$. Here, $f_{lastval}$ was computed for each spectrum separately, based on a universal (constant for all spectra) limiting value of $A$: $A_{lim} = 0.05$. That is, $f_{lastval}$ is the highest frequency for which $A > A_{lim}$. For all $f > f_{lastval}$, a high-frequency tail of the form $E_0(f) \sim f^{-4}$ was added after Kaihatu et al. (2007) by extrapolating the trend from the last $n = 10$ reliably estimated $E_0(f)$ values (yellow line in **Fig. 3**):

$$E_0(f) = \tilde{E}_0 f^{-4} \qquad \text{for } f > f_{lastval}, \tag{8}$$

where:

$$\tilde{E}_0 = \sum_{j=0}^{n-1} E_0\left(f_{lastval-j}\right) f_{lastval-j}^{-4} \Big/ \sum_{j=0}^{n-1} f_{lastval-j}^{-8}. \tag{9}$$

### 3.3 Mean wave parameters

In calculation of mean (integral) wave parameters, frequencies $f < f_{min} = 0.04$ Hz (corresponding to wave periods higher than 25 s) were ignored. This limit corresponds to the approximate boundary between wind-generated and infragravity waves, as well as to the lower frequency limit typically used in spectral wave models (e.g., Holthuijsen, 2007). Thus, the mean wave





parameters were computed for $f_{\min} < f < f_{\max}$. In the final dataset, two sets of those parameters are provided, referred to as observational one (for $f_{\max} = f_{\text{lastval}}$) and modelled one (for $f_{\max} = \infty$). The spectral moments $m_n$ of $E_0(f)$ are defined as:

$$m_n = \int_{f_{\min}}^{f_{\text{lastval}}} E_0(f)f^n df + C \frac{1}{3-n}\tilde{E}_0 f_{\text{lastval}}^{n-3} \qquad \text{for } n \in \mathbb{N}, \qquad (10)$$

where $\tilde{E}_0$ is computed from (9), $C = 0$ if $f_{\max} = f_{\text{lastval}}$ and $C = 1$ if $f_{\max} = \infty$. Based on $m_n$, the following wave parameters are calculated: the significant wave height $H_s$, the mean absolute wave period $T_{m0,1}$, the mean absolute zero-crossing period $T_{m0,2}$, and the so-called energy period $T_{m-1,0}$:

$$H_s = 4\sqrt{m_0}, \qquad (11)$$

$$T_{m0,1} = m_0/m_1, \qquad (12)$$


$$T_{m0,2} = \sqrt{m_0/m_2}, \qquad (13)$$

$$T_{m-1,0} = m_{-1}/m_0. \qquad (14)$$

### 3.4 Output data

There are two output files per each deployment with rows representing bursts. The first one ('DepID_properties.txt') contains

the information on burst (number and time), mean water depth $\overline{z_{\text{lf}}}$, $f_{\text{lastval}}$, and the four mean wave parameters defined in Eqs. (11–14), in two versions, i.e., for $C = 0$ and $C = 1$, respectively, in formula (10). The second file provides wave energy spectra for frequencies from 0.040039 to 0.5 Hz with step $\Delta f = \frac{1}{1024}$ Hz (472 columns). **Fig. 4** provides a visualisation of an example one-month period of data. **Table 2** provides the dataset content (Swirad et al., 2023).

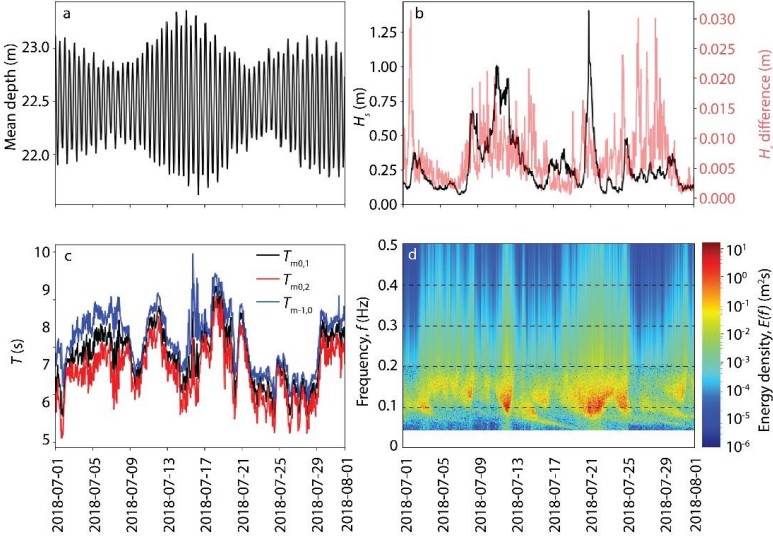

**Figure 4: An example of outputs for one month (2018-07) of deployment HBK9: (a) mean depth $\overline{z_{\text{lf}}}$; (b) primary y-axis: significant wave height, $H_s$ for $f_{\max} = \infty$, secondary y-axis: the difference between $H_s$ for $f_{\max} = \infty$ and for $f_{\max} = f_{\text{lastval}}$; (c) wave period, $T$ for $f_{\max} = \infty$, (d) wave energy spectra $E_0(f)$.**





**Table 2: Dataset content. 'DepID' stands for deployment ID.**

| File name | Number of files | Type | Rows | Columns |
|---|---|---|---|---|
| DepID.txt | 24 | input | Single measurements at 1 Hz frequency (full seconds) in 1024-element bursts (hh:00:00 to hh:17:03) starting at full hours UTC | 1. Time ['yyyy-mm-dd hh:mm:ss']<br>2. Burst ID [1:n]<br>3. Measured pressure (dbar) |
| airpressure.txt | 1 | input | Hourly measurements starting 2013-07-21 00:00:00 UTC | 1. Atmospheric pressure at the sea level (mbar) |
| bursts2waves.py | 1 | code | n/a | n/a |
| DepID_properties.txt | 24 | output | Single bursts | 1. Burst ID [1:n]<br>2. Time ['yyyy-mm-dd hh:mm:ss']<br>3. Mean depth $\overline{z_{lf}}$ (m)<br>4. $f_{lastval}$ (Hz)<br>5. $H_s$ (m) for $f_{max} = f_{lastval}$<br>6. $T_{m0,1}$ (s) for $f_{max} = f_{lastval}$<br>7. $T_{m0,2}$ (s) for $f_{max} = f_{lastval}$<br>8. $T_{m-1,0}$ (s) for $f_{max} = f_{lastval}$<br>9. $H_s$ (m) for $f_{max} = \infty$<br>10. $T_{m0,1}$ (s) for $f_{max} = \infty$<br>11. $T_{m0,2}$ (s) for $f_{max} = \infty$<br>12. $T_{m-1,0}$ (s) for $f_{max} = \infty$ |
| DepID_spectra.txt | 24 | output | Single bursts | 1-472. Wave energy density, $E(f)$ (m$^2$s) at 0.040039 to 0.5 Hz with 1/1024 Hz step |

### 3.5 Quality control

The instruments remained at the sea bottom thanks to the anchor weight. However, a few times they were moved by ice or strong waves resulting in an abrupt change in mean depth visible in the output data (e.g. **Fig. 5a**). This situation happened

three times: in VES1 bursts #83 (depth rise of ~1 m) and #370 (depth drop of ~2.3 m), and in GAS5 burst #13420 (depth rise of ~0.7 m). In the case of VES1 burst #83 and GAS5 burst #13420 it happened in between bursts with no impact on calculated wave energy spectra and bulk parameters. Therefore, we left the output unchanged. If the dataset is used for tide analysis, timeseries should be split at the depth change event and treated separately. To identify erroneous bursts, we looked at the energy density for $f < 0.5$ Hz and identified two bursts with abnormally high energy density at low frequencies that resulted

in erroneous calculation of bulk parameters (e.g. **Fig 5b**): VES1 burst #370 and HBK1 burst #44. In the first case the error resulted from instrument displacement during the burst. In the second case mean depth rised by ~0.5 m, remained higher for a few hours and dropped back to a typical level. There was no anomaly in atmospheric pressure and we speculate that the artefact may be due to a presence of glacier ice at the sea surface. In both cases we replaced all output wave parameters with *NaN*.

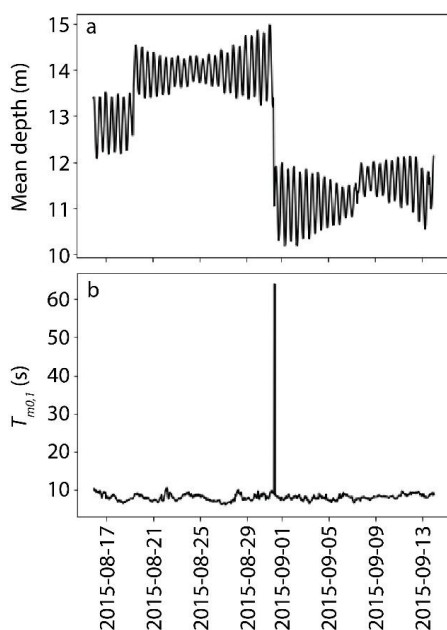

**Figure 5: An example of data errors for deployment VES1: (a) mean depth $\overline{z_{\mathrm{lf}}}$; (b) mean wave period $T_{\mathrm{m0,1}}$ for $f < 0.5$ Hz.**

## 4 Results

For all bays except Rettkvalbogen timeseries length exceeded one year providing information on seasonal variability in wind wave conditions. The largest waves characterise Veslebogen, a western-most of the analysed northern bays (**Fig. 6**). Mean full dataset $H_{\mathrm{s}}$ ranged from 0.25 m in eastern Isbjørnhamna to 0.43 m in Veslebogen and respective 99[th] percentile $H_{\mathrm{s}}$ equalled

1.21 m and 1.96 m. Waves were the highest in the first and last quarter of the year with the highest mean $H_{\mathrm{s}}$ of 0.53 m in October-December and 99[th] percentile $H_{\mathrm{s}}$ of 2.32 m in January-March, both in Veslebogen (**Table 3**). A seasonal trend is also clearly visible in **Fig. 7**. Winter months are characterised by generally higher and longer waves, a finding consistent with the multi-decadal wave model reanalysis of Wojtysiak et al. (2018) for open Greenland Sea, west of Hornsund.



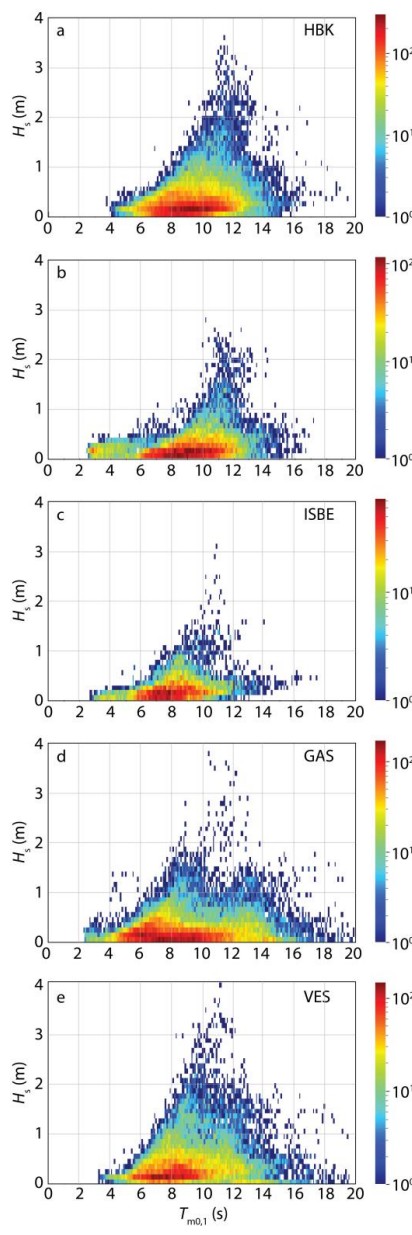


**Figure 6: Distribution of significant wave height, $H_s$ (y-axix; range: 0-4 m with 0.1 m bins) and mean absolute wave period, $T_{m0,1}$ (x-axis; range 0-20 s with 0.1 s bins) with $f_{max} = \infty$ for (a) Hansbukta (HBK), (b) western Isbjørnhamna (ISBW), (c) eastern Isbjørnhamna (ISBE), (d) Gåshamna (GAS), and (e) Veslebogen (VES).**



**Table 3: Summary of significant wave height, $H_s$: mean, 99th percentile for the full dataset and by quarters of the year, and mean full dataset wave period: mean absolute wave period, $T_{m0,1}$, mean absolute zero-crossing period, $T_{m0,2}$, and energy period, $T_{m-1,0}$. HBK = Hansbukta, ISB = Isbjørnhamna (W = western, E = eastern), GAS = Gåshamna, VES = Veslebogen. Rettkavbogen (RET) is excluded as the 13-day duration is not sufficient to derive seasonal statistics.**

|  | HBK | ISBW | ISBE | GAS | VES |
|---|---|---|---|---|---|
| Mean $H_s$ (m) | 0.33 | 0.26 | 0.25 | 0.26 | 0.43 |
| 99th percentile $H_s$ (m) | 1.71 | 1.5 | 1.21 | 1.33 | 1.96 |
| Jan-Mar mean $H_s$ (m) | 0.44 | 0.34 | 0.33 | 0.35 | 0.5 |
| Jan-Mar 99th percentile $H_s$ (m) | 2.06 | 1.76 | 1.5 | 1.54 | 2.32 |
| Apr-Jun mean $H_s$ (m) | 0.21 | 0.13 | 0.16 | 0.2 | 0.35 |
| Apr-Jun 99th percentile $H_s$ (m) | 0.97 | 0.48 | 0.72 | 1.18 | 1.71 |
| Jul-Sep mean $H_s$ (m) | 0.23 | 0.16 | 0.17 | 0.17 | 0.35 |
| Jul-Sep 99th percentile $H_s$ (m) | 1.03 | 0.82 | 1 | 0.79 | 1.48 |
| Oct-Dec mean $H_s$ (m) | 0.43 | 0.4 | 0.35 | 0.34 | 0.53 |
| Oct-Dec 99th percentile $H_s$ (m) | 1.96 | 1.91 | 1.25 | 1.47 | 2.25 |
| Mean $T_{m0,1}$ (s) | 9.51 | 9.2 | 8.31 | 8.84 | 9.19 |
| Mean $T_{m0,2}$ (s) | 8.72 | 8.23 | 7.33 | 7.93 | 8.39 |
| Mean $T_{m-1,0}$ (s) | 10.36 | 10.34 | 9.5 | 9.93 | 10.09 |

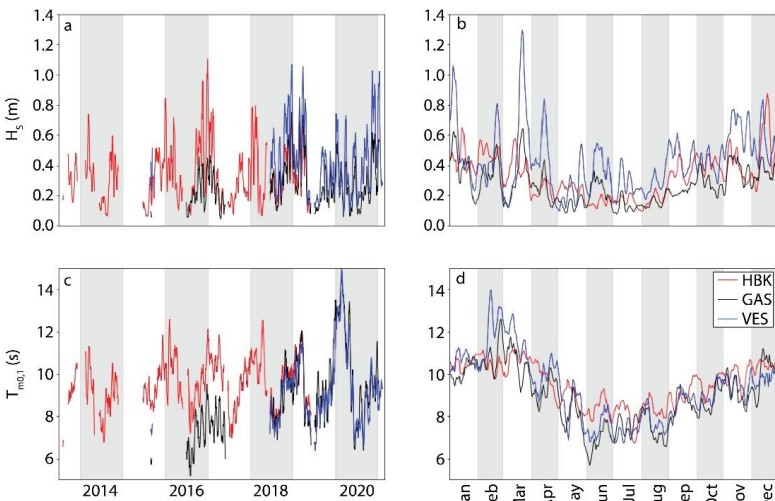


**Figure 7: Summary of the wind wave characteristics for Hansbukta (HBK), Gåshamna (GAS) and Veslebogen (VES) for $f_{max} = \infty$: (a) mean daily significant wave height, $H_s$ smoothed with a 15-day moving average; (b) mean daily significant wave height, $H_s$ for days of year smoothed with a 5-day moving average; (c) mean daily absolute wave period, $T_{m0,1}$ smoothed with a 15-day moving average; (d) mean daily absolute wave period, $T_{m0,1}$ for days of year smoothed with a 5-day moving average.**

**5 Data availability**

The inputs, outputs and the Python code described in this manuscript are available in the PANGAEA repository (https://doi.org/10.1594/PANGAEA.954020; Swirad et al., 2023). Raw data downloaded from the instruments are part of the IG PAS LONGHORN oceanographic monitoring and they are available at the IG PAS Data Portal (https://doi.org/10.25171/InstGeoph_PAS_IGData_NBP_2022_005; Swirad et al., 2022). As the monitoring program is on-

going, future raw and processed data will be uploaded to the IG PAS Data Portal (https://dataportal.igf.edu.pl/).



## 6 Summary

We present the first multi-year continuous wind wave and water level dataset for Hornsund fjord, Svalbard. 24 single deployments of underwater RBR sensors at 8–23 m depth between July 2013 and February 2021 were used to measure water levels in five bays of northern (Hansbukta, western Isbjørnhamna, eastern Isbjørnhamna, Rettkvalbogen, Veslebogen) and one of southern (Gåshamna) Hornsund. Raw data (Swirad et al., 2022) were subsampled to 1024 s sets (~bursts) at 1 Hz measurement interval at 1 h burst interval that were then used to derive mean water levels, wave spectra and bulk wave parameters. We describe the procedure (available also as a Python code) that includes subtracting atmospheric pressure, depth calculation, Fast Fourier Transform, correction for the decrease of the wave orbital motion with depth and adding a high-frequency tail. We performed quality control on the output data. The dataset can be used to e.g. characterise wind wave climate in Hornsund, identify seasonal to inter-annual trends, calibrate and validate wave models, and facilitate e.g. analysis of sea ice impact on wave attenuation, empirical modelling of wave run-up on Arctic beaches and predicting future change.

**Author contributions.** MM initiated and maintains the oceanographic monitoring in Hornsund. ZMS secured the funding. ZMS wrote the code and processed the data with the support from AH and MM. All authors wrote the manuscript.

**Competing interests.** The authors declare that they have no conflict of interest.

**Acknowledgements.** We thank Kacper Wojtysiak for sharing his MATLAB code, Aleksandra Stępień and Adam Słucki (HańczaTECH) for helping in the underwater work, and the Polar Polish Station crew for maintaining the oceanographic and meteorological monitoring.

**Financial support.** This study was funded by National Science Centre of Poland (grant no. 2021/40/C/ST10/00146). Acquisition of raw data (Swirad et al., 2022) was funded by National Science Centre of Poland (grant no. 2013/09/B/ST10/04141), IG PAS LONGHORN oceanographic monitoring in collaboration with Polish Polar Station Hornsund, and the Ministry of Education and Science of Poland (statutory activities no. 3841/E-41/S/2022).

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
