# Peer review of "Wind wave and water level dataset for Hornsund, Svalbard (2013-2021)"

_Earth System Science Data, 2023_

## Author Response (AR1)

Below we provide responses to reviewers' comments. Line numbers refer to the modified clean manuscript.

**RC1: Anonymous Referee #1**

The manuscript is overall well written, and the dataset is of interest to the scientific community I therefore recommend publication after some corrections.

Thank you.

The introduction is good but, in my opinion, doesn't sell the dataset as well as it might deserve. There is a lot of discussion about the types of models that are primarily used which always include a lot of acronyms etc. Could this perhaps be rewritten slightly to better highlight the usefulness of the data, or a bit more clearly? I'm aware that this goes beyond the scientific content of the paper, but it would be nice to see the data exploited.

We justify the references to different wave models and to studies based on them by the usefulness of our dataset in spectral wave modelling. As we write in the text, wave models at high latitudes generally suffer from insufficient validation with observational data. We expect wave modelers to be among the readers and, although we are aware that the acronyms might be difficult to follow for readers less familiar with wave modeling jargon, we decided to leave that part of the introduction unchanged.

As for the last comment: this is a data description paper, and we follow the rules of ESSD in that we do not include any data interpretation etc. A subset of the data described here has been used in Herman et al. (2019) which is an example of how our data, combined with modeling, can be applied to an analysis of wave conditions in an Arctic fjord. We have added a note on that in the introduction (lines 54-55) and summary (line 268) that read:

*Notably, the study used a subset of the dataset described in this paper to validate the wave spectral model (Herman et al., 2019).*

And

*...calibrate and validate wave models (as shown by Herman et al., 2019)...*

Why is there a gap in data during winter? Were the instruments recovered then? Or was there a higher failure rate for sensors during winter? This could be clarified.

Thanks for this comment. We now clarify it in lines 104-107 that read:

*Initially the deployments were short (< 100 days) and usually restricted to the fieldwork season (late spring to autumn). Since 2015, however, deployments were typically ~1-year long with instrument recovery and re-deployment during summer field campaigns. As a result of the COVID-19 pandemics, it was impossible to recover instruments in summer 2020, and the last two deployments (GAS5 and VES3) were > 550 days long and ended with the battery death.*

Chapter 5. Will this data be provided in some of these databases processed in the same way?

Yes. We now clarify it in lines 257-258 that read:

*As the monitoring program is on-going, future raw and processed in the same way data will be uploaded to the IG PAS Data Portal (https://dataportal.igf.edu.pl/).*

**Comments**

R8. The record is continuous or near continuous please add that.

Done

R9. Would a better term for mean water levels be "sea surface height", SSH?

The term "sea surface height" is used in oceanography (and especially in satellite ocean altimetry) to describe the sea level relative to a given geoid. As we do not attempt to relate our measurements to any fixed, global frame of reference, we think that the "mean water level" is more appropriate. We have not changed that.

R12. "Wind induced waves"

The term "wind waves" for wind-generated ocean surface waves is widely used and well established (used in book titles, encyclopedia entries, etc). We do not have any doubts about using it throughout our paper (including the title). We have not changed that.

R18. Information -> data

Done

R20. In the Svalbard

Done

R22. Is storminess the best word?

We have replaced it with 'increasing frequency and strength of storms'.

R23. Larger -> higher

Done

R30. Move the term hourly to earlier in the sentence. "Hourly, e.g. significant"

Done

R31. "ERA-40 reanalysis data"

Done

R32. Swell vs sea?

We have replaced it with 'seasonal trends in swell and wind sea'.

R33. Calculate an average

Done

R56. Please provide a position already here

We are unsure which position the reviewer refers to, but we have added lat/long information (line 61).

R98. Bottom -> bed

Done

R184. Per -> for

Done

R198. Bottom -> bed

Done

R198. Moved -> transported

Done

R199, 201. Happened -> occurred

Done

R202. Therefore, the data is left unchanged.

Done

R203. Looked at -> investigated

Done

**RC2: Tsubasa Kodaira**

**General comments**

This manuscript presents a dataset comprising 24 deployments of underwater RBR sensors at depths ranging from 8 to 23 meters, conducted between July 2013 and February 2021. These sensors were utilized to measure water levels in five bays of Hornsund, Svalbard. The manuscript details the process of generating the dataset of wind waves and water levels from the original time series. This dataset holds significance due to its uniqueness, usefulness, and completeness.

Thank you.

However, the authors should enhance the method of data analysis for wind waves, particularly in clarifying the calculation of wave power spectrum. Describing the analysis method as "standard way" may not provide readers with sufficient information to understand the data. Details such as data length, window size, and tapering can be described.

Thank you for this comment. It is very important to stress that our dataset includes not only the derived wave energy spectra, but the individual bursts with pressure time series as well, so that anyone wishing to use the data can test their own analysis methods (e.g. an alternative to FFT based on wavelets), experiment with different algorithm parameters, analyze nonlinear effects (our analysis is based on the linear wave theory), and so on. The choice of "the best" method may depend on a particular application

(e.g. the wave characteristics of interest). We also provide our processing scripts with the dataset, allowing each user to repeat and, if necessary, modify each step of the analysis. We have added a comment in the summary (lines 270-271) that read:

*We provide individual bursts with pressure times series and the code for the users to apply different analysis methods, use alternative algorithm parameters, analyse nonlinear effects, etc. depending on the application.*

We agree that the wording "in a standard way" is a bit unfortunate. We removed this phrase from the revised text and we added references to the literature at the beginning of section 3.2 (lines 129-132) that read:

*The deployment files were imported into Spyder (Python 3.9) and processed on the burst-by-burst basis, with an algorithm described below (see also Wang et al., 1986, Karimpour et al., 2017, Marino et al., 2022, and references therein). Importantly, all steps described below are based on the linear wave theory; alternative data processing methods (e.g., Bonneton et al., 2018) might be applied to the original burst data to capture nonlinear effects, but they are not considered here.*

As for details of our algorithm, we provide information on the data length = 1024 (section 3.1 and in Table 2, now also in section 3.2). The Python fft function with default settings was used to compute the spectra, and no windowing was applied.

The publication is recommended after implementing these corrections.

 **Specific comments:**

L22: Please provide more specific information rather than just citing the IPCC report.

We now provide more detail in lines 21-23 that read:

*Continuous wave observations in the coastal Arctic are needed to better understand how i) decreasing sea-ice extent – pan-Arctic annual mean extent decrease of 3.5-4.1% per decade (IPCC, 2019) or 1.5 to 3-fold increase of the length of sea-ice free season along pan-Arctic coasts (Barnhart et al., 2014) between 1979 and 2012...*

L43: Is "total wave energy" the summation of both kinetic and potential energy, or solely potential energy? Additionally, it would be beneficial to specify the type of data employed for the evaluation and the definition of wave period.

We replaced "total wave energy" with "significant wave height" and "wave period" with "mean absolute wave period" (line 52).

We clarify that a portion of this dataset was used by a modelling study of Herman et al. (2019) to demonstrate utility of this dataset for model evaluation (lines 54-55):

*Notably, the study used a subset of the dataset described in this paper to validate the wave spectral model (Herman et al., 2019).*

L48-L54: Consider improving the sentences here, as their logical flow is currently challenging to follow.

We have split the information on large-scale (now lines 28-47) and local models (now lines 49-55) with clarification of the latter (see next comment).

L50: What notable information or details were added by Herman et al. (2019)?

We now clarify it in lines 52-55 that read:

*The study added a considerable detail into wind wave transformation in the nearshore environment of Hornsund by including fjord bathymetry, which allowed resolving depth-induced wave breaking and bottom friction on wind conditions.*

L70: Could you clarify the meaning of "UTM33x"?

We now clarify it in Fig. 1 caption (lines 75-76) as:

*Universal Transverse Mercator coordinate system zone 33X (UTM33X)*

In Table 1 we replaced UTM coordinates with lat/long (see next comment).

L105 (Table 1): Instead of utilizing X and Y, why not employ GPS positions?

That's a good point. We have now changed UTM coordinates (meters) to lat/long coordinates (degrees).

L161: The determination of "f_lastval" should depend on the specific method used for calculating the PSD. Please address this concern by providing clarification on the spectral analysis performed in this study, including spectral uncertainty.

We do not agree. The $f_{lastval}$ does not depend on the details of calculation of energy spectra. There are several ways of computing this cutoff (the simplest one amounts to throwing away all wavelengths shorter than $2z_s$, where $z_s$ is the mean sensor depth), but as long as the *linear* wave theory is used, $A$ is independent of wave amplitude (see Eq. (7) in our paper) and thus independent on the wave spectrum. In other words, $A$ reflects the rate of attenuation of the wave signal with depth and has nothing to do with the amplitude of the waves at the sea surface. We have added a corresponding comment to the revised text of section 3.2 (lines 177-179):

*(note that, consistent with the linear wave theory used throughout this analysis, the values of A depend only on water depth and frequency of a given spectral component, but not on the amplitude of that component).*

L168: What precisely are the "mean wave parameters"?

The exact parameters (significant wave height, mean absolute wave period, mean absolute zero-crossing period and so-called energy period) are listed at the end of this section. We have not made any changes.

L176: Kindly verify the meaning of "zero-crossing period," as it appears to pertain to the mean wave period calculated from time series analysis.

Throughout this paper, we use terminology established in spectral wind wave modelling and related fields, because researchers working on these subjects are the most likely readers of our paper and users of our data. In particular, we use the term mean wave parameters (equivalent of bulk wave parameters,

or integral wave parameters; see e.g. the first sentence in section 3.3) for quantities characterizing the whole energy spectrum (as opposed to individual spectral components). Analogously, the zero-crossing period is defined based on the spectral moments, as in formula (13) – see, e.g., Holthuijsen (2007). In the last case, the situation is analogous to that of the significant wave height: it can be computed in a "traditional" way from time series of sea surface elevation, but the definition (11) based on the zeroth spectral moment is much more widely used, as it can be computed from both observations and results of spectral modelling. We have not made any changes.